# Meier–Gorlin Syndrome: Clinical Misdiagnosis, Genetic Testing and Functional Analysis of *ORC6* Mutations and the Development of a Prenatal Test

**DOI:** 10.3390/ijms23169234

**Published:** 2022-08-17

**Authors:** Maria S. Nazarenko, Iuliia V. Viakhireva, Mikhail Y. Skoblov, Elena V. Soloveva, Aleksei A. Sleptcov, Ludmila P. Nazarenko

**Affiliations:** 1Research Institute of Medical Genetics, Tomsk National Research Medical Center, Russian Academy of Sciences, 634050 Tomsk, Russia; 2Department of Medical Genetics, Siberian State Medical University, 634050 Tomsk, Russia; 3Research Centre for Medical Genetics, 115522 Moscow, Russia

**Keywords:** Meier–Gorlin syndrome, Jeune syndrome, *ORC6*, exon skipping variant, prenatal genetic testing

## Abstract

Meier–Gorlin syndrome (MGS) is a rare genetic developmental disorder that causes primordial proportional dwarfism, microtia, the absence of or hypoplastic patellae and other skeletal anomalies. Skeletal symptoms overlapping with other syndromes make MGS difficult to diagnose clinically. We describe a 3-year-old boy with short stature, recurrent respiratory infections, short-rib dysplasia, tower head and facial dysmorphisms who was admitted to the Tomsk Genetic Clinic to verify a clinical diagnosis of Jeune syndrome. Clinical exome sequencing revealed two variants (compound heterozygosity) in the *ORC6* gene: c.2T>C(p.Met1Thr) and c.449+5G>A. In silico analysis showed the pathogenicity of these two mutations and predicted a decrease in donor splicing site strength for c.449+5G>A. An in vitro minigene assay indicated that variant c.449+5G>A causes complete skipping of exon 4 in the *ORC6* gene. The parents requested urgent prenatal testing for MGS for the next pregnancy, but it ended in a miscarriage. Our results may help prevent MGS misdiagnosis in the future. We also performed in silico and functional analyses of *ORC6* mutations and developed a restriction fragment length polymorphism and haplotype-based short-tandem-repeat assay for prenatal genetic testing for MGS. These findings should elucidate MGS etiology and improve the quality of genetic counselling for affected families.

## 1. Introduction

Meier–Gorlin syndrome (MGS) is an osteodysplastic syndrome that causes primordial proportional dwarfism [1,2,3,4,5]. MGS prevalence is estimated to be less than 1–9 cases per 1,000,000 people; however, the true prevalence is difficult to ascertain because of the phenotypic heterogeneity [1]. Microtia and aplastic or hypoplastic patella—in addition to growth retardation—constitute the three core clinical findings in MGS. Nonetheless, the typical diagnostic triad is not always present at diagnosis [1].

Other possible signs and symptoms are neonatal progeroid appearance, feeding and respiratory problems during infancy, early infantile epileptic encephalopathy, congenital cardiac anomaly, hearing loss, microcephaly, craniosynostosis, ichthyosis, lipodystrophy, adrenal insufficiency, various bone defects including femoral asymmetry, coxa valga, abnormal ribs, delayed bone age and urogenital anomalies, such as cryptorchidism in males and mammary hypoplasia in post-pubertal females [1,6,7,8,9,10,11]. Patients with MGS may also have characteristic facial features including a narrow nose, microstomia, full lips and micrognathia [1,12]. Most patients with this syndrome have normal intelligence, but motor and/or language development is delayed [1].

Some of these defects are thought to result from cell-type-specific DNA replication defects during development. In patients with MGS, mutations in a dozen genes of the pre-replication complex (*ORC1*, *ORC4*, *ORC6*, *CDT1*, *CDC6*, *MCM5*, *MCM3*, *MCM7*, *GMNN*, *CDC45*, *DONSON* and *GINS2*) have been detected, which can also be present in DNA replication-associated genetic diseases [13,14].

The diagnosis of MGS is complicated by high clinical variability and considerable genetic heterogeneity without a clear genotype–phenotype correlation. Here, we report a case of MGS misdiagnosed as Jeune syndrome because of short stature, short limbs and a severe thoracic anomaly. Using sequencing technologies, we identified two known MGS-associated mutations—c.2T>C(p.Met1Thr) and c.449+5G>A in the *ORC6* gene—in a couple with two pregnancy losses. In silico prediction and functional analysis showed pathogenicity and a possible mechanism of action of these two mutation sites. Finally, we developed a prenatal genetic testing system for *ORC6*-based MGS with the help of restriction fragment length polymorphism (RFLP) and haplotype-based short tandem repeat (STR) analysis.

## 2. Results

### 2.1. Case Presentation

The patient was admitted to the Tomsk Genetic Clinic when he was 3 years old. From this point on, he presented with short stature, high forehead, tower head, downslanted palpebral fissures, narrow nose, micrognathia, bilateral microtia, short neck, short limbs, narrow chest, micromelia, brachydactyly and muscular hypotonia (Figure 1B). He had normal developmental milestones and intellect. Echocardiography showed no signs of heart defects. Chest radiography detected short horizontal ribs. Moderate internal hydrocephalus and enlargement of subarachnoid spaces of the frontal lobes were revealed by magnetic resonance imaging of the brain. He received gastrostomy feeding to optimize nutrition.

The boy was born of the second pregnancy (Figure 1A). At 12 weeks of gestation, a high risk of chromosomal anomalies in the fetus was revealed. Karyotype analysis results were normal (46, XY). Ultrasonography at 20 weeks showed ventricular enlargement and intrauterine growth retardation. The boy was born at 34 weeks of gestation via caesarean section with a birth weight of 1.6 kg and length 41 cm. Feeding problems, respiratory distress syndrome and neonatal pneumonia were registered.

He is of Slavic descent, and his family history includes Turner syndrome (in a paternal cousin), bronchial asthma and arterial hypertension (in maternal relatives). His parents are nonconsanguineous. The first pregnancy of his mother ended in a miscarriage at 25 weeks of gestation. Ultrasonography at 19 weeks of gestation raised concerns about shortened fetal long bones. The third pregnancy also ended in a miscarriage, at 12 weeks of gestation.

According to clinical exome sequencing, the patient is a compound heterozygote for two mutations in the *ORC6* gene: a c.2T>C transition within the initial methionine of the transcription start site (p.Met1Thr) and a splice site mutation (c.449+5G>A) (Figure 1C). Sanger sequencing validation confirmed the presence of these variants and inheritance from parents. Mutations c.2T>C (p.Met1Thr) and c.449+5G>A were inherited from the father and mother, respectively (Figure 1D). Although these variants have been described earlier, mechanisms of pathogenicity remain quite interesting to study and hypothesize [5].

### 2.2. In Silico Analysis of Mutations c.2T>C(p.Met1Thr) and c.449+5G>A of ORC6

The mutations c.2T>C(p.Met1Thr) and c.449+5G>A of the *ORC6* gene in a compound heterozygous state have been relatively common among patients with MGS since de Munnik et al. (2012) identified similar mutations in four patients [5].

Genetic variant c.2T>C(p.Met1Thr) is classified as pathogenic in VarSome (ACMG: PVS1, PP5 and PM2). This variant is present in a population database (rs146795505, gnomAD European (non-Finnish) population, minor allele frequency (MAF) = 0.06%) and a Russian medical database (RUSeq, MAF = 0.04%). ClinVar contains an entry for this variant (variation ID: 253272). Genetic variant c.2T>C(p.Met1Thr) affects the initiator methionine of ORC6 mRNA. In ORC6 mRNA, there are alternative methionine codons (Met20, Met50 and Met59) that may be used for translation initiation in subjects with a mutation in the first AUG codon. The ORC6 protein is known to have various isoforms, some of which have a putative later start position (e.g., the H3BT22 potential isoform that is computationally mapped in the UniProt database [15], and several isoforms are described in the GTEx database [16] and Ensembl [17]) and thus do not encompass the mutated c.2T>C or c.1A>G base pair that can be present in patients with MGS. We hypothesized that disease-causing start codon variants activate downstream translation initiation and result in an N-terminally truncated protein. Further research is needed to fully determine functional significance of variant c.2T>C(p.Met1Thr) of the *ORC6* gene.

The second variant, c.449+5G>A, is located in intron 4 of the *ORC6* gene. It is classified as likely pathogenic in VarSome (ACMG: PP3, PM2 and PP5). This variant is present in a population database (rs572314014, gnomAD European (non-Finnish) population, MAF = 0.03%) and a Russian medical database (RUSeq, MAF = 0.01%). ClinVar contains an entry for this variant (variation ID: 253273). According to analysis using bioinformatic tool NNSplice, *ORC6* exon 4 was assumed to have a weak 5′ or 3′ splice site (NNSplice score 0.67 and 0.79, respectively). We annotated the c.449+5G>A mutation by means of the Human Splicing Finder software. This mutation causes a loss of a canonical splice donor site (HSF Donor site = 82.72% to 73.16% (−11.56%)).

Analysis of the *ORC6* exon 4 splice sites using MaxEntScan indicated that exon 4 contains a weak 3′ splice site with a maximum entropy score of 5.60, but a relatively strong 5′ splice site (score 7.33), which is weakened (to score 0.76) by the c.449+5G>A mutation. varSEAK confirms this finding (score + 33.81% to −38.84% (−72.65%)). It is well known that exons with weak splice sites are especially sensitive to a shifted balance between positive and negative splicing regulatory elements. Thus, we hypothesized that variant c.449+5G>A leads to splicing changes in *ORC6* exon 4 and lower amounts of the full-length ORC6 protein via a shifted balance between positive and negative splicing regulatory elements.

### 2.3. In Vitro Minigene Assays of the c.449+5G>A Mutation in the ORC6 Gene

To validate the predictions of the in silico splice site tools, an in vitro minigene assay was performed on variant c.449+5G>A of the *ORC6* gene. To create a wild type (WT) minigene, we amplified *ORC6* exon 4 with adjacent introns and cloned it into a minigene vector. A construct carrying the c.449+5G>A variant was created by site-directed mutagenesis on a WT template (Figure 2A).

Each construct was transfected into HEK293T cells. After 48 h, the cells were lysed, RNA from the cells was extracted and reverse transcribed, and PCR analysis was performed. Gel electrophoresis revealed that the chimeric transcript from the mutant minigene matches in length the transcript from the empty vector, and that the chimeric transcript from the WT construct is rather long (Figure 2B). Sanger sequencing confirmed that the WT construct includes exon 4, and in the mutant construct, we observed only plasmid exons, meaning complete exon 4 skipping (Figure 2C).

### 2.4. A Prenatal Genetic Diagnostic System

The mother of the patient asked for prenatal diagnosis for her next (third) pregnancy, which at the time was at gestational 6 weeks. We urgently developed a molecular genetic testing system involving the analysis of both mutations of the *ORC6* gene and three additional linked STR markers. The results of PCR-RFLP and polyacrylamide gel electrophoresis (PAGE) analyses of the family are displayed in Figure 3A,B.

A 387 bp fragment normally containing two natural *Fat*I restriction sites (yielding DNA fragments of 172, 158 and 57 bp) was obtained for the first variant (Figure 3A). Mutation c.2T>C (p.Met1Thr) produces a loss of one restriction site (yielding DNA fragments of 172 and 215 bp). The PCR-RFLP results indicated that the proband and his father had this mutation (Figure 3A).

Amplification-created restriction sites were used for the detection of the c.449+5G>A mutation (Figure 3B). Forward primers contained a mismatch creating the restriction site for the *Taq*I enzyme. In the case of the mutation, the 84 bp amplicon broke up into DNA fragments of 64 and 20 bp during the restriction digestion analyses. Mutation c.449+5G>A was confirmed in the proband and his mother (Figure 3B).

Our data suggested that the new method allows to identify mutations c.2T>C (p.Met1Thr) and c.449+5G>A of the *ORC6* gene in the carrier parents and in the compound heterozygous state in the affected child.

Family haplotypes of three STRs linked to the *ORC6* gene are presented in Table 1. Dinucleotide repeat (AC)_n_ (STR1), tetranucleotide repeat (AGAT)_n_ (STR2) flanking the *ORC6* gene at a distance not exceeding 0.5 Mbp, and intragenic STR (AGAT)_n_ of exon 7 were used. For the maternal chromosome, all three STRs were informative. The father’s STR1 was uninformative. An example of PCR and DNA fragment analysis of AGAT repeats in exon 7 of the *ORC6* gene is presented in Figure 3C.

As a result, a reliable prenatal genetic testing system for ORC6-based MGS was created involving direct analysis of mutations c.2T>C(p.Met1Thr) and c.449+5G>A in combination with haplotype-based three-STR analysis. Unfortunately, the mother had a miscarriage at 12 weeks of pregnancy, and the requested prenatal genetic testing was not carried out.

## 3. Discussion

Genetic variants of *ORC6* are associated with autosomal recessive MGS (Figure 4). The *ORC6* gene is located in chromosomal region 16q11.2. According to gnomAD, the LoF o/e (loss-of-function observed/expected) score for *ORC6* is 0.84 [18]. The loss of one copy of *ORC6* is likely not lethal in humans. There are ~500 genetic variants of *ORC6* in gnomAD [19]. Missense (32%) and intronic (32%) genetic variants are the most prevalent according to gnomAD.

More than 90 genetic variants in *ORC6* are registered in the ClinVar database [20]. Most of them (>50) are classified as variants of uncertain significance. Eleven pathogenic and likely pathogenic variants are listed in ClinVar and PubMed NCBI (Figure 4). These variants are located mainly in the N-terminal domain of ORC6. Only two pathogenic splicing variants are found in the middle (M) domain. The C-terminal domain of ORC6 contains pathogenic frame shift and missense variants.

Bicknell et al. (2011) showed that MGS can result from a loss of ORC6 function [4]. Homozygosity for the c.602_605delAGAA mutation (or c.598_601AGAA [1] (p.Lys201fs) in ClinVar) in the *ORC6* gene is lethal for humans [2]. The fetal phenotype of MGS includes severe intrauterine growth retardation, dislocation of knees, gracile bones, clubfeet, micrognathia and small thorax (Figure 4).

It is noteworthy that Li et al. (2017) identified homozygous mutation c.67A>G (p.Lys23Glu) with complete uniparental disomy of chromosome 16 in an 11-year-old Chinese boy with MGS [3]. The boy has short stature, microtia, small patella, slender body build, craniofacial anomalies and small testes with a gonadotropin level within the reference range.

Bicknell et al. (2011) found three affected children from one family with compound heterozygous mutations in the *ORC6* gene: one loss-of-function mutation caused by a 2 bp deletion designated as c.257_258delTT (p.Phe86fs) and missense mutation c.695A>C inducing a substitution of an amino acid (p.Tyr232Ser) [4]. All three patients manifested growth retardation, microtia and other ear malformations, aplastic or hypoplastic patella, full lips and micrognathia; two males had cryptorchidism, and a female had mammary hypoplasia (Figure 4).

In the largest study in this field (45 persons with MGS), seven patients were found to carry mutations in the *ORC6* gene [5]; three patients had genotype c.257_258delTT (p.Phe86fs) + c.695A>C (p.Tyr232Ser); and the combination of mutations c.2T>C(p.Met1Thr) + c.449+5G>A in the *ORC6* gene was detected in four patients (Figure 4). Two out of the above seven patients did not exhibit all the traits of the classic triad. One patient had normal stature but small ears and absent patellae. Another one had microtia and short stature without patellar aplasia/hypoplasia. Six out of the seven patients with *ORC6* mutations had patellar malformations. They also manifested intrauterine growth retardation, and one of them had overt microcephaly. Malformed ears were noted in three patients. All seven patients in question had micrognathia, and six of them had malformed genitalia. In terms of skeletal anomalies, delayed bone age was registered in four patients, and contractures/clubfeet in two patients. No cardiac aberrations were detected in the children. Respiratory and feeding problems during infancy were identified in four and five patients, respectively. All patients had normal intellect, but three and two of them showed delayed motor and speech development, respectively.

Owing to the diverse clinical features of MGS even in patients with similar mutations in *ORC6*, the diagnosis of this rare disease can be challenging, pointing to the need for in-depth genetic investigation. By clinical exome sequencing, we were able to detect the compound heterozygous mutations c.2T>C(p.Met1Thr) and c.449+5G>A of the *ORC6* gene in the 3-year-old boy who was admitted to the Tomsk Genetic Clinic for verifying the clinical diagnosis of Jeune syndrome (Figure 1). Genetic variant c.2T>C(p.Met1Thr) is classified as pathogenic in VarSome. The second likely pathogenic variant, c.449+5G>A, is located in intron 4 of the *ORC6* gene.

Because of the location of c.449+5G>A, i.e., at a canonical splice donor site, we further investigated the matter using in silico and molecular strategies. The Human Splicing Finder 3.1 software (GENOMNIS SAS, Marseille, France) suggested a loss of a canonical splice donor site. NNSplice, MES and varSEAK indicated a higher chance of splicing changes of exon 4 with the mutant allele than with the WT sequence. The ideal material for researching the mechanism by which c.449+5G>A affects splicing is a biological sample from a patient. Although ORC6 performs functions in many cellular processes, it is expressed rather weakly in many tissues; accordingly, we decided to utilize a common method of splicing analysis: a minigene assay. Using this assay, we were able to demonstrate an exon 4 skipping in a specific region surrounding the splicing site by means of the DNA sample from the proband (Figure 2). The data indicate probable exon removal during mRNA processing.

Exon 4 length is 90 nucleotides; hence, its elimination will lead to an in-frame deletion in the protein. Thus, at the protein level, exon 4 skipping generates a missense mutation changing serine to arginine at position 120, and the protein shortens by 30 amino acid residues (p.Ser120Arg; 121Tyr_150Lysdel). The deleted region is a part of the M domain of the ORC6 protein. The M domain contains a long DNA-binding motif that is necessary for DNA replication [21]. Nevertheless, all functional research to date has been performed on point mutations, some of which are located near the predicted deletion [22]. These mutations impair chromosome–protein binding. It is likely that the deletion under study has a similar effect, affecting only this function. On the other hand, the 30-amino-acid deletion shortens the protein significantly, and therefore the mutated ORC6 can become unstable and completely lose its function. Both hypotheses require further experimental confirmation.

We describe a case of ORC6-based MGS misdiagnosed as Jeune syndrome because of short stature, short limbs and a severe thoracic anomaly (Table 2). On the other hand, short stature and microtia are recognized as major hallmarks of MGS that have been reported in many cases of this disease including the present case [1,3,5]. A patellar defect was not noted in the present case, similarly to the case reported by de Munnik et al. [5]. It should be mentioned that neither radiological nor ultrasonographic analysis of patellae was used in our case. Although tower head and moderate internal hydrocephalus with enlargement of subarachnoid spaces of the frontal lobes were identified in the present case, they have not been reported in patients with ORC6-based MGS before.

The prenatal diagnostic system was developed for our case of ORC6-based MGS (Figure 3). In addition to mutations c.2T>C(p.Met1Thr) and c.449+5G>A of the *ORC6* gene, the assay performs indirect detection of polymorphic STRs linked to this gene. PCR-RFLP and PAGE analyses of the two mutations of *ORC6* confirmed the sequencing results. The prenatal genetic diagnostic system was developed urgently because the patient’s mother was already pregnant. In this regard, there was a risk that a direct test for mutations may cause problems, and there was no time for redesign. Therefore, to be on the safe side, an indirect diagnostic assay of STRs linked to *ORC6* was created. STR markers have their own risks because they may be uninformative. Nevertheless, the dinucleotide repeat (AC)_n_ and tetranucleotide repeat (AGAT)_n_ that were found to flank the *ORC6* gene at a distance not exceeding 0.5 Mbp were informative, as was intragenic repeat (AGAT)_n_ in exon 7. The presented system has a reliability margin and can reveal potential artefacts of prenatal diagnosis, in particular, maternal contamination. We believe that the prenatal diagnostic assay for ORC6-based MGS can be useful, even though it was not applied in practice owing to the miscarriage.

On the other hand, one study [2] suggests that mothers of patients with MGS may experience miscarriages. The mother of our patient with MGS also had two pregnancy losses. We expect that preimplantation genetic testing will become the first-line option instead of prenatal genetic testing for such families. The newly developed STR analyses can be employed for preimplantation genetic testing of MGS caused by *ORC6* gene mutations.

It should also be mentioned that in our case, *ORC6* gene mutations were found by sequencing in a scientific department of our institution. Later, the parents applied for prenatal diagnosis to the clinical department. PCR/RFLP is the most optimal method for prenatal diagnosis of known mutations. This technique is easier, faster and cheaper than Sanger sequencing especially in small departments with low workload.

Clinical features of *Drosophila* with *Orc6* mutations (DmOrc6) overlap with those of people with MGS [22,24]. Flies carrying the p.Tyr225Ser mutation in DmOrc6 die at the third-instar larval stage and have severe replication and developmental defects. The surviving mutant flies are short with a long head, rough eye spots, missing posterior scutellar bristles and irregular disoriented hairs on the abdomen and are flightless [24].

To study the functions of Orc6, Chesnokov et al. (2020) managed to create a humanized Orc6-based *Drosophila* model of MGS [22]. The chimeric Orc6 protein contains the human N-terminal domain and the *Drosophila* C terminus and rescues Orc6-deficient flies by yielding viable adults phenotypically indistinguishable from WT animals. Those researchers showed that two known MGS-associated variants in Orc6 are situated in different functional domains of the protein and result in either impaired DNA binding (p.Lys23Glu) by Orc6 or a loss of the protein associated with the core ORC (p.Tyr232Ser).

The flies carrying Orc6-p.Lys23Glu often are able to develop to adulthood [22]. In that study, defective scutellar bristles and inability of mutant flies to fly were registered in Orc6-null flies rescued with Orc6-p.Lys23Glu. It is known that the sensory bristles and flight muscles of adult *Drosophila* undergo very rapid and complex morphogenesis. Such cells require more active protein synthesis to accommodate their rapid rate of development. Mutations in the *Orc6* gene can give rise to a cell-type-specific DNA replication defect and a decrease in protein synthesis during rapid critical developmental stages.

Patients with ORC6-based MGS possess diverse clinical features. We think that a mouse model can be more informative for functional analysis of human missense and splicing mutations. A humanized mouse model can also be helpful for testing therapeutic agents that target the missplicing defect. There is no Orc6-based murine model of MGS at present.

ORC6 is a part of the origin recognition complex (ORC). This protein is crucial for the initiation of DNA replication in the cell, whether for human, yeast or fruit fly [25,26,27]. ORC6 is highly conserved among eukaryotes, and consequently studies on model organisms provide deeper insight into the function of this protein. Some reports suggest that *Drosophila* Orc6 mutations and the humanized Orc6-based *Drosophila* model cause minor chromosomal defects in larval neuroblasts [22,28]. During mitosis, ORC6 localizes to kinetochores and a reticulum-like structure around the cell periphery. Hence, it is implicated in chromosome replication and segregation with cytokinesis both in *Drosophila* cells and in human cells [28,29].

The mutations found in patients with MGS cause a spectrum of cellular phenotypes that partially overlap with the effect of MCM mutations, including impaired licensing, altered S-phase progression and proliferation defects [30]. Notably absent from this list of phenotypes is chromosomal instability or an increased predisposition to cancer. We assume that some individuals with MGS are at an increased risk of cancer, but this notion is currently not supported by the clinical record because MGS is such a rare disorder. Therefore, patient-specific cell lines and an Orc6-based mouse model of MGS should be useful for testing genome instability and the increased predisposition to cancer.

Additionally, cilia develop from centrosomes/centrioles, and small-interfering-RNA-mediated silencing of human *ORC6* causes centrosome aberrations and impaired ciliogenesis in fibroblasts as well as an aberrant chondroinduction phenotype [31]. Knockdown experiments on Orc6 in *Danio rerio* have also resulted in cilium shortening; Orc6 deficiency in *D. rerio* gives rise to a phenotype that includes growth retardation, left–right asymmetry abnormalities and cystic kidney formation, typical of cilia dysfunction [32]. The centrosome is critical for promoting the switch from asymmetric to symmetric neuroepithelium cell division during prenatal neurogenesis in mammals, and a failure of this process can induce expansion defects of the pool of neuronal progenitor cells as well as microcephaly and dwarfism [33,34,35]. We can theorize that mutations of the *ORC6* gene can influence the balance of cell divisions via centrosome aberrations.

It is known that cilia dysfunction may contribute to the clinical manifestation of ORC-deficient MGS, including ORC6-deficient MGS [32]. Several reports from human case studies and ciliopathic animal models strongly indicate that craniofacial development and skull malformation are associated with defective cilia formation either directly or indirectly, through several signaling pathways [36,37,38].

We constructed a map of functional and physical protein associations using genetic variants of our patient with a predicted pathogenicity A and B category by means of SOPHiA DD^TM^ and STRING [39] (Appendix A). ORC6 proved to not be connected with other proteins. Sixteen proteins (GLI3, MKKS, SUFU, HYDIN, CCDC40, TCTE1, MCHR1, BBS9, PCDH15, GUCY2D, CRX, C2orf71, RP1L1, CACNA1F, VCAN and ARFGEF2) were annotated as a ‘Cilium’ cellular component or were found to be associated with ‘Ciliopathies’ (GO:0005929, WP4803, GOCC:0005929, and KW-1186; FDR < 0.05; Appendix A). Variants in genes *GLI3*, *SUFU*, *BBS9*, *MKKS* and *ARFGEF2* have been described in subjects with craniofacial phenotypes (Appendix A) [40,41,42,43,44,45,46,47,48,49].

According to the ACMG (Varsome), heterozygous rare variants of uncertain significance c.2179G>A (rs121917710, p.Gly727Arg) in the *GLI3* gene and c.3046G>C (rs149644732, p.Gly1016Arg) in the *ARFGEF2* gene were identified in our patient (Appendix A). Variants near or in the *ARFGEF2* gene are implicated in early-life head circumference but not skull defects [47,48,49].

There is a report about an association between p.Gly727Arg in *GLI3* and digital abnormalities [50]. Nonetheless, there is evidence from mouse models regarding an influence of the Gli3 protein on skull defects through cilia and Gli-mediated sonic hedgehog (SH) signaling during development [37,51,52]. These data probably indicate the involvement of missense mutation p.Gly727Arg of the *GLI3* gene in tower head during skull morphogenesis of our patient. We believe that this variant of the protein may be inefficient when transducing SH signaling in the case of cilia dysfunction due to *ORC6* gene mutations. A connection between SH signaling and DNA replication has been found in relation to cancer initiation [53]. Further investigation is needed to clarify the links among cilia, signaling pathways and DNA replication in terms of embryonic development.

Therefore, research on both human cell lines in vitro and humanized *Drosophila* and mouse models will provide much-needed insight into the functional consequences of *ORC6* mutations. Clarifying the molecular mechanisms underlying MGS is essential for designing new treatments for the patients. If there is a reduction in splicing efficiency of *ORC6* mRNA owing to the disturbed interplay of splicing enhancers and silencers, it is possible to manipulate them to rescue the splicing and increase the amounts of the protein. Consequently, it is important to create patient-specific cell lines and a mouse Orc6-based model of MGS for testing of therapeutic agents.

## 4. Materials and Methods

### 4.1. Study Design

This is a case report of clinical features and of next-generation sequencing of the patient’s nucleic acids. The patient presented with recurrent respiratory infections, short-rib dysplasia and short stature. The patient’s mother and father were also examined to confirm the findings.

### 4.2. The Diagnostic Genetic Testing

#### 4.2.1. Genomic-DNA Preparation

Peripheral-blood samples from the patient and his parents were collected into EDTA-containing tubes. The patient’s genomic DNA was extracted from peripheral-blood leucocytes with the phenol–chloroform DNA extraction procedure [54]. DNA concentration and purity of the samples were determined on an Invitrogen Qubit 4 Fluorometer (Thermo Fisher Scientific, Waltham, MA, USA) and NanoDrop 2000 spectrophotometer (Thermo Fisher Scientific), respectively. The DNA samples were stored at 4 °C prior to use.

#### 4.2.2. Library Preparation, Clinical Exome Sequencing and Bioinformatic Analysis

DNA libraries of the patient were prepared from 200 ng of DNA using the SOPHiA Clinical Exome Solution according to the manufacturer’s recommendations (SOPHiA GENETICS, Rolle, Switzerland). Next-generation sequencing (NGS) was performed on the NextSeq-500 Platform (Illumina, San Diego, CA, USA), and Sophia DDM algorithms were used for the bioinformatic analysis of the data. The Clinical Exome Solution spans 12 Mbp of the target region covering more than 4490 genes with known inherited disease-causing mutations.

#### 4.2.3. PCR and Sanger Sequencing

To confirm the suspected pathogenicity of the found variants, validation was implemented through PCR combined with bidirectional Sanger sequencing. Primers targeting the mutation site were designed for PCR amplification (Thermo Fisher Scientific) with sequencing (ORC6_ex1F: 5′-CGTCCTGTCCTAACCAATCC-3′ and ORC6_ex1R: 5′-TGAAGTAGGCCCTAAACCCC-3′; ORC6_ex4F: 5′-TGGTTGATACTTTCAGTCTCTTTTC-3′ and ORC6_ex4R: 5′-TTAGAAGTATTCCTTGGGGATG-3′). The DNA samples were sequenced with the BigDye Terminator v3.1 cycle sequencing kit on an Applied Biosystems 3730 Genetic Analyzer (Thermo Fisher Scientific). The results were interpreted with the help of Chromas 2.6.3 software (Technelysium, South Brisbane, QLD, Australia).

### 4.3. In Silico Analysis

The following databases were utilized for annotating the genetic variants: (i) allele frequency: gnomAD [55] and RUSeq [56]; (ii) pathogenicity prediction: VarSome [57]; (iii) splicing prediction: NNSplice, MaxEntScan (MES), Human Splicing Finder (HSF) and varSEAK [58,59,60,61]; (iv) medical relevance: ClinVar [62]; and (v) analyses of protein–protein interaction networks and of functional enrichment: STRING [39].

### 4.4. The In Vitro Minigene Assay

#### 4.4.1. Cell Culture

A human embryonic kidney 293 cell line (HEK293T) was cultured in DMEM supplemented with 1% of a penicillin–streptomycin solution (PanEco, Moscow, Russia) and with 10% of fetal bovine serum (Biosera, Nuaillé, France) at 37 °C in a humidified atmosphere containing 5% of CO_2_.

#### 4.4.2. Minigene Constructs

The genomic locus covering *ORC6* exon 4 with flanking introns was amplified from a control DNA sample with the following primers: ORC6 F (5′-caccagaattctggagctcgagATCCATTGTCCTGTTCTGGTTAATTTTCAG-3′) and ORC6 R (5′-gggatcaccagatatctgggatccCATGTGAAGACTGTTTTAATGACAGATATCATCTCT-3′). Vector pSpl3-Flu2-TKdel was linearized using restriction digestion at *Xho*I and *Bam*HI sites. *ORC6* PCR products were cloned into the pSpl3-Flu2-TKdel vector by means of the Gibson Assembly Master Mix (NEB, Ipswich, MA, USA). Thus, we obtained the WT construct. A plasmid carrying the c.449+5G>A variant was created using the Phusion Site-Directed Mutagenesis Kit (Thermo Fisher Scientific), following the manufacturer’s recommendations. Briefly, the WT vector was amplified with the following primers: ORC6 mut f (/Pi/-5′-CATGCAAGTAGATATTTCATTAA-3′) and ORC6 mut r (/Pi/-5′-CTGAAAGCAGTGCAGCAGAAG-3′). This amplicon was digested with *Dpn*I and then ligated by means of the T4 ligase. The ligated product was transformed into *Escherichia coli* (XL10 Gold strain) with the standard heat-shock method. After the transformation, single clones were sequenced, and a clone harboring the desired variant was chosen for further experiments.

#### 4.4.3. The Splicing Assay

The WT and Mut plasmids were transfected into HEK293T cells by the calcium phosphate method. The cells were seeded at 6 × 10^4^ cells/well in 24-well poly-L-lysine-coated plates with an antibiotic-free medium 24 h prior to transfection. We transfected 500 ng of plasmid DNA per well. At 48 h after the transfection, total RNA was isolated using the ExtractRNA reagent (Evrogen, Moscow, Russia), following the manufacturer’s instructions. The RNA was treated with DNase I (Thermo Fisher Scientific) and reverse transcribed with MMuLV H-reverse transcriptase (Dialat, Moscow, Russia) using random hexanucleotides. To analyze the structure of the resultant chimeric transcript, we employed primers for plasmid exons: TurboFP F (5′-ACAAAGAGACCTACGTCGAGCA-3′) and GFP R (5′-AGCTCGATCAGCACGGGCACGAT-3′). PCR products were separated by 2% agarose gel electrophoresis. All amplicons were sequenced to verify the correctness of the obtained transcript.

### 4.5. The Prenatal Genetic Diagnostic System

#### 4.5.1. PCR-RFLP and PAGE

PCR was carried out in a 20 μL reaction mixture, which consisted of 0.9 U of Taq polymerase (SibEnzyme, Novosibirsk, Russia), a buffer for Taq polymerases (SibEnzyme, Novosibirsk, Russia), water (molecular biology grade), 1.5 mM MgCl_2_, 0.13 mM each dNTP, 6% of dimethyl sulphoxide, 1 pmol of appropriate oligonucleotide primers per reaction and a 10 ng DNA sample. Two amplification programs were used, with annealing at either 55 or 60 °C.

We chose the ‘60 °C’ PCR program and suitable primers (F: 5′-AGACCCTGATTGGTTTGTGG-3′ and R: 5′-ACGGCCGAGAAGTTTTTCTT-3′) for the amplification of the c.2T>C (p.Met1Thr) mutation site. A 387 bp fragment normally containing two natural *Fat*I restriction sites (yielding DNA fragments of 172, 158 and 57 bp) was obtained. Mutation produces a loss of one restriction site (yielding DNA fragments of 172 and 215 bp).

Amplification-created restriction sites were used for the detection of the c.449+5G>A mutation. Forward primers contained a mismatch creating the restriction site for the *Taq*I enzyme. The amplification with suitable primers (F: 5′-ACTGCTTTCAGCATGCAAGTCG-3′ and R: 5′-TTCCTTGGGGATGAAAAACC-3′) was conducted according to thermal-cycling program ’55 °C’. In the case of the mutation, the 84 bp amplicon yielded DNA fragments of 64 and 20 bp during the restriction digestions.

The restriction analysis was performed in a 20 μL reaction mixture, according to the manufacturer’s instructions (SibEnzyme, Novosibirsk, Russia). Restriction analysis results were examined by PAGE in 7% gels. The gels were stained with the GelRed dye (Biotinum, Fremont, CA, USA) and documented by means of ultraviolet light on a Gel Doc XR+ System transilluminator (Bio-Rad, Hercules, CA, USA).

#### 4.5.2. PCR and Fragment Analysis

Three polymorphic STR markers were selected for indirect genetic testing. Dinucleotide repeat (AC)_n_ (STR1), tetranucleotide repeat (AGAT)_n_ (STR2) flanking the *ORC6* gene at a distance not exceeding 0.5 Mbp and intragenic STR (AGAT)_n_ of exon 7 were used. We carried out PCR with fluorescently labeled forward or reverse primers to analyze the STRs. The following primers were applied: exon 7 (AGAT) F: 5′-TGATGGTGCATGCCTGTAAT-3′ and R: 5′-R6G-AATGGCTCCAATTCCGTTTA-3′, STR1 (AC) F: 5′-FAM-GTTTTCTCCCGCATGACAAG-3′ and R: 5′-CAAATTGAGATGAACATTTTGG-3′, and STR2 (AGAT) F: 5′-TTTCCCCAGCTGGAATTAAA-3′ and R: 5′-FAM-AGCCTGGGCAACAAAGTTAG-3′. Exon 7 (AGAT) was amplified by means of standard program ’60 °C’, and STR1 and STR2 by program ‘55 °C’ with 0.6 M betaine instead of dimethyl sulphoxide in the reaction. The PCR results were previewed by PAGE and detected by fragment analysis on an Applied Biosystems 3130xl Genetic Analyzer (Thermo Fisher Scientific). The data were analyzed in the GeneMapper software.

## 5. Conclusions

Our study offers a genetic diagnostic assay for MGS. In our case report, this disease was found to be clinically misdiagnosed as Jeune syndrome. This is the first functional validation of an effect of splice site variant (c.449+5G>A) of the *ORC6* gene by a human-cell-based minigene assay. Our data clearly confirmed an exon 4 skipping mechanism for the *ORC6* transcript. To our knowledge, this is the first haplotype-based STR assay for prenatal genetic testing for ORC6-based MGS. More studies are needed to evaluate potential effects of *ORC6* gene variants on multiple cellular processes in in vitro human-cell experiments, in the humanized *Drosophila* model and in a prospective humanized mouse model.

## Figures and Tables

**Figure 1 ijms-23-09234-f001:**
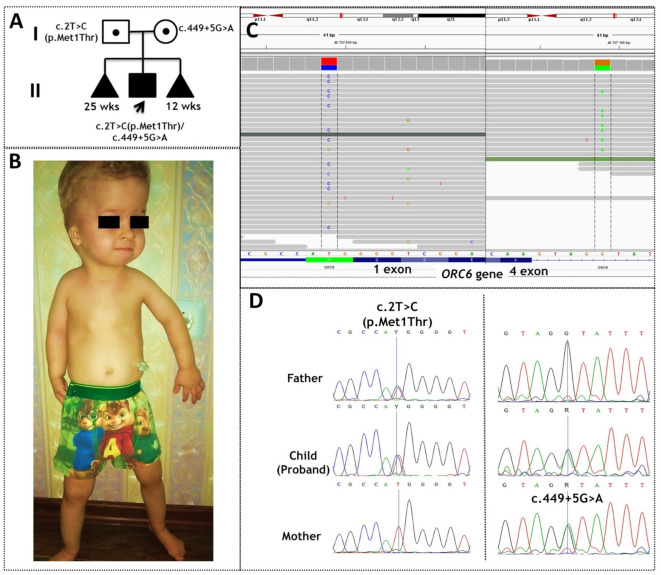
The case of Meier–Gorlin syndrome. (**A**) The pedigree of the proband; (**B**) the phenotype of the proband; (**C**) molecular genetic testing of the patient by clinical exome sequencing; (**D**) Sanger sequencing validation of the variants and of their inheritance from the parents.

**Figure 2 ijms-23-09234-f002:**
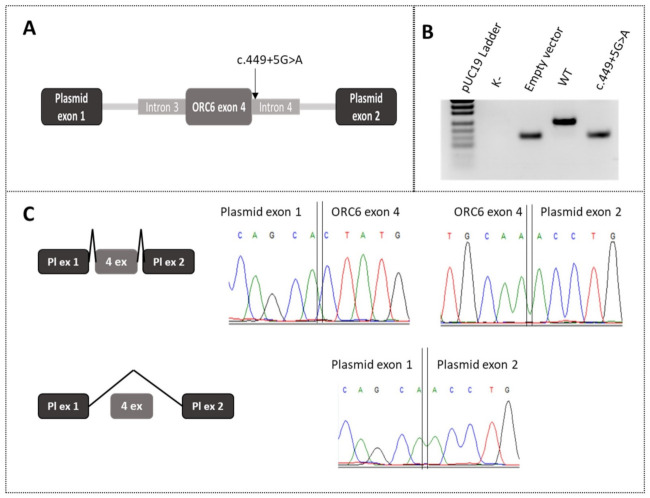
In vitro minigene analysis of c.449+5G>A of the *ORC6* gene. (**A**) The scheme of plasmid construction for the minigene assay; (**B**) an agarose gel image from RT-PCR analysis of chimeric transcripts; (**C**) results of Sanger sequencing of splicing products in minigenes.

**Figure 3 ijms-23-09234-f003:**
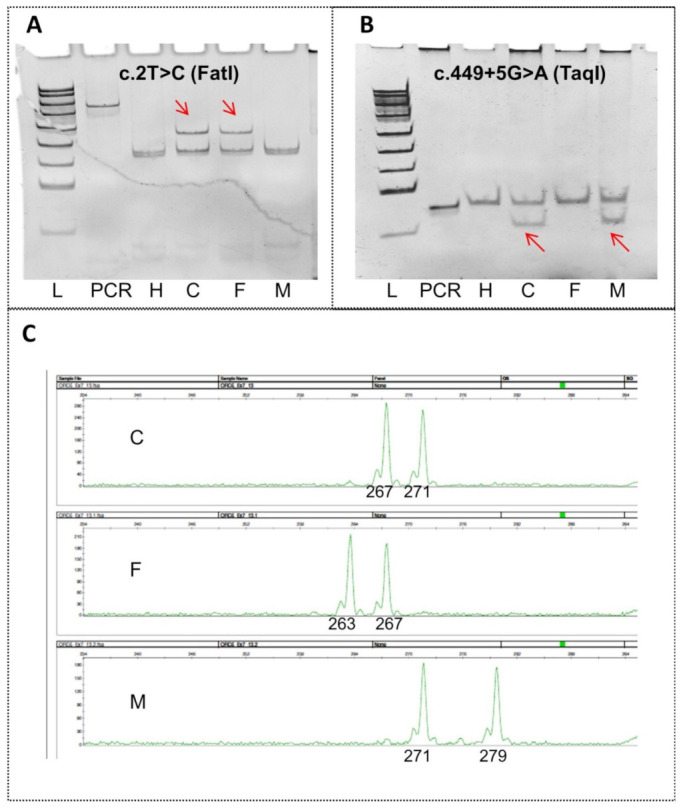
The prenatal genetic diagnostic system for c.2T>C(p.Met1Thr) and c.449+5G>A of the *ORC6* gene. (**A**,**B**) PCR-RFLP and PAGE analyses; (**C**) PCR and fragment analysis of AGAT repeats of exon 7. Lane L: pUC19/MspI Ladder; lane ‘PCR’: the undigested amplicon; lane H: an unrelated healthy control; lane C: the affected child; lane F: father; and lane M: mother.

**Figure 4 ijms-23-09234-f004:**
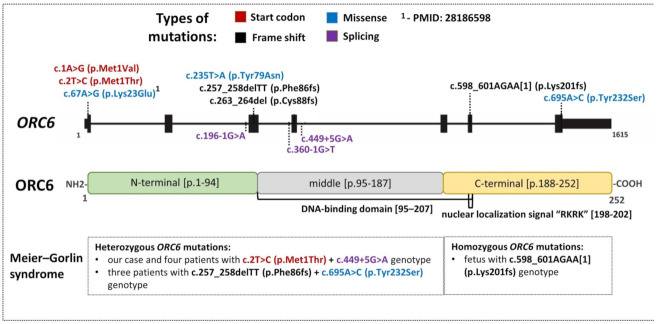
The *ORC6* gene and protein and phenotypes of mutations [2,4,5].

**Table 1 ijms-23-09234-t001:** Family haplotypes of STRs linked to the *ORC6* gene.

Locus, Distance from Gene, Repeat		Child (Proband) ^1^	Father ^1^	Mother ^1^
STR1	0.01 Mbp, (AC)_n_	**308**/**310**	**308**/308	308/**310**
*ORC6*	mutations	**c.2T>C**/**c.449+5G>A**	**c.2T>C**/N	N/**c.449+5G>A**
	Exon 7, (AGAT)_n_	**267** **/** **271**	**267**/263	279/**271**
STR2	0.52 Mbp, (AGAT)_n_	**99**/**103**	**99**/90	107/**103**

^1^ The alleles of mutant chromosomes are boldfaced.

**Table 2 ijms-23-09234-t002:** Possible symptoms of ORC6-based MGS and Jeune syndrome.

Features	ORC6-Based MGS		Jeune Syndrome
	Our Patient ^1^ (Male, 3 y.o.)	7 Patients ^1^ (5 Males and 2 Females; 3 y 10 m to 15 y 5 m) [5]	13 Patients ^1^ (8 Males and 5 Females; 5 Weeks to 22 y.o.) [23]
Main skeletal features	**intrauterine growth retardation;** **short stature;** tower head; short neck; **short limbs;** **narrow chest;** brachydactyly.	**intrauterine growth retardation;** **short stature;** patellar aplasia/hypoplasia; **slender ribs** and long bones; microcephaly; clinodactyly; contractures/clubfeet.	**short stature;** **short limbs;** **short-rib dysplasia with narrow chest (persistent respiratory manifestations);** unusually shaped pelvis; extra fingers and/or toes.
Facial features (change with age)	**bilateral microtia;** **high forehead;** **downslanted palpebral fissures;** narrow nose; **micrognathia.**	**bilateral microtia;** malformed ears; **high forehead;** **downslanted palpebral fissures;** **micrognathism** with full lips and small mouth; accentuated nasolabial folds; convex nasal profile.	no.
Additional clinical features	**respiratory and feeding problems during infancy and gastrostomy;** **normal intellect;** muscular hypotonia; moderate internal hydrocephalus; enlargement of subarachnoid spaces of frontal lobes.	**respiratory and feeding problems during infancy**; nasogastric feeding/**gastrostomy**; **normal intellect** but delayed motor or speech development; abnormal genitalia (cryptorchidism or small testes/mammary hypoplasia or hypoplastic labia minora/majora).	renal insufficiency; abnormality of retinal pigmentation; abnormality of the liver.

^1^ Overlapping clinical features between ORC6-based MGS and Jeune syndrome are boldfaced.

## Data Availability

Raw data are available from the corresponding author upon request.

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
