# Peer review of "Meier–Gorlin Syndrome: Clinical Misdiagnosis, Genetic Testing and Functional Analysis of ORC6 Mutations and the Development of a Prenatal Test"

_ijms, 2022, doi:10.3390/ijms23169234_

Round 1

Reviewer 1 Report

In the manuscript “Meier-Gorlin Syndrome: Clinical Misdiagnosis as Jeune Syndrome, Genetic Testing and Functional Analysis of ORC6 Mutations and the Development of a Prenatal Test” authors present new clinical case of the Meier-Gorlin Syndrome (MGS). MGS is a rare genetical developmental disorder characterized by high clinical variability therefore a thorough analysis of any new case would help to improve the medical practice and clinical diagnosis.

Overall, manuscript provides valuable information for clinical diagnostics and molecular basis underlying MGS based on ORC6 mutations. In the revised manuscript authors did a good job addressing the comments of the reviewers. With all significant comments addressed manuscript can now be recommended for the publication.

Author Response

Author response:  We appreciate the time and effort that you dedicated to providing feedback on our manuscript. Thank you!

Reviewer 2 Report

This article is suitable for the topic of the special issue.

The authors have made changes and additions that answer almost all my questions.

I still have two comments:

1) If the authors are so interested in the differential diagnosis olnly with Jeune syndrome, then I suggest leaving this in the discussion, but removing it from the title.

2)Regarding the absence of a patellar defect (lines 275-277), it is still unclear how its absence was confirmed, if no research was conducted on them (X-ray, ultrasound, CT).

Using the example of one case of an extremely rare hereditary disease, the paper presents the possibilities of clinical, molecular genetic and scientific methods for diagnostics and prenatal diagnostics. Therefore, this work can be presented in a special issue.

Author Response

Reviewer 2’s comment:  If the authors are so interested in the differential diagnosis only with Jeune syndrome, then I suggest leaving this in the discussion, but removing it from the title.

Author response: We agree with the reviewer’s advice and have therefore changed the title (page 1, lines 1-4). The new title is “Meier-Gorlin Syndrome: Clinical Misdiagnosis, Genetic Testing and Functional Analysis of ORC6 Mutations and the Development of a Prenatal Test”.

Reviewer 2’s comment:  Regarding the absence of a patellar defect (lines 275-277), it is still unclear how its absence was confirmed, if no research was conducted on them (X-ray, ultrasound, CT).

Author response:  Thank you for pointing this out. We wrote, “A patellar defect was not noted in the present case, similarly to the case reported by de Munnik et al. [5]. It should be mentioned that neither radiological nor ultrasonographic analysis of patellae was used in our case.” Primary patellar ossification typically starts at age 5 to 6 years [PMID: 30664715]. So, patellar defects are hard to detect in children younger than 5 years, especially without imaging techniques. Our three-year-old boy seems to be the youngest patient among the very few liveborn cases of ORC6-based MGS. Proband’s patellae should be assessed with care at a later age.